# Histone Deacetylase Inhibitor, Trichostatin A, Synergistically Enhances Paclitaxel-Induced Cytotoxicity in Urothelial Carcinoma Cells by Suppressing the ERK Pathway

**DOI:** 10.3390/ijms20051162

**Published:** 2019-03-07

**Authors:** Fu-Shun Hsu, June-Tai Wu, Jing-Yi Lin, Shao-Ping Yang, Kuan-Lin Kuo, Wei-Chou Lin, Chung-Sheng Shi, Po-Ming Chow, Shih-Ming Liao, Chun-I Pan, Jo-Yu Hong, Hong-Chiang Chang, Kuo-How Huang

**Affiliations:** 1Department of Urology, New Taipei City Hospital, New Taipei City 112, Taiwan; fs_hsu@outlook.com (F.-S.H.); emilyang719@gmail.com (S.-P.Y.); antibody0123@gmail.com (K.-L.K.); sanguine444@gmail.com (S.-M.L.) vickypan0686@gmail.com (C.-I.P.); 2Graduate Institute of Clinical Medicine, College of Medicine, National Taiwan University, Taipei 106, Taiwan; 3Department of Dermatology, National Taiwan University Hospital, Taipei 100, Taiwan; junetai.wu@gmail.com; 4Department of Dermatology, Chang Gung Memorial Hospital, Keelung 204, Taiwan; lindajylin@cgmh.org.tw; 5College of Medicine, Chang Gung University, Taoyuan 333, Taiwan; 6Graduate Institute of Toxicology, College of Medicine, National Taiwan University, Taipei 106, Taiwan; 7Department of Urology, National Taiwan University Hospital, Taipei 100, Taiwan; meow1812@gmail.com (P.-M.C.); yvonnehorng@gmail.com (J.-Y.H.); changhong@ntu.edu.tw (H.-C.C.); 8Department of Pathology, National Taiwan University Hospital, Taipei 100, Taiwan; weichou8@ms52.hinet.net; 9Graduate Institute of Clinical Medical Sciences, College of Medicine, Chang Gung University, Taoyuan 333, Taiwan; csshi@mail.cgu.edu.tw

**Keywords:** urothelial carcinoma, trichostatin A, histone deacetylase inhibitor, chemotherapy, paclitaxel

## Abstract

Trichostatin A (TSA), an antifungal antibiotic derived from *Streptomyces*, inhibits mammalian histone deacetylases, and especially, selectively inhibits class I and II histone deacetylase (HDAC) families of enzymes. TSA reportedly elicits an antiproliferative response in multifarious tumors. This study investigated the antitumor effects of TSA alone and in combination with paclitaxel when applied to two high-grade urothelial carcinoma (UC) cell lines (BFTC-905 and BFTC-909). Fluorescence-activated cell sorting, flow cytometry, and 3-(4,5-dimethylthiazol-2-yl)-2,5-diphenyl tetrazolium assay were used to assess TSA’s cytotoxicity and effects on apoptosis induction. TSA induced synergistic cytotoxicity, when combined with paclitaxel (combination index < 1), resulted in concomitant suppression of paclitaxel-induced activation of phospho-extracellular signal-regulated kinase (ERK) 1/2. A xenograft nude mouse model confirmed that TSA enhances the antitumor effects of paclitaxel. These findings demonstrate that the administration of TSA in combination with paclitaxel elicits a synergistic cytotoxic response. The results of this study indicate that the chemoresistance of UC could be circumvented by combining HDAC inhibitors to target the ERK pathway.

## 1. Introduction

Paclitaxel was first derived in 1967 from the bark of trees of the genus, *Taxus*; its antitumor activity has been proven by numerous pharmacological and clinical studies [1]. Bladder urothelial carcinoma (UC) is the 4th most common form of cancer in men and the 11th most common among women in the United States. Upper urinary tract UC (UTUC) accounts for 20–30% of all UC cases in Taiwan [2]. Following radical surgery, 20–50% of cases still face recurrence and metastases. Cisplatin-based chemotherapy is the primary treatment regime for bladder UC that has progressed to the metastatic phase. Even after treatment with paclitaxel, cisplatin/gemcitabine, or methotrexate/vinblastine doxorubicin/cisplatin, the overall response rate is only 50–60%, and the side effects can negatively affect patients. Recent promising developments in immunotherapy, including inhibitors of programmed cell death proteins, PD-1 and PD-L1, have proven effective in 15–29% of patients with metastatic UC following recurrence after platinum-containing chemotherapy drugs, and the side effects have been generally tolerable [3,4,5]. Nearly all cases of metastatic UC develop drug resistance and eventually lead to death. Novel compounds are urgently required to enhance treatment effectiveness in terms of overcoming drug resistance and reducing the side effects of chemotherapy.

Human cancers typically exhibit aberrant epigenetic modifications to the structure of chromatins without alterations to the DNA sequence. The primary protein molecules of chromatins are histones, which are subject to several types of posttranslational modifications that alter the interactions between DNA and histones and exert subsequent effects on gene transcription, DNA repair, and DNA replication [6,7]. Histone acetylation and deacetylation play critical roles in gene expression and translational activation [8,9]. Gene transcription is associated with the induction of histone acetylation by histone acetyl transferases; however, the enzymatic removal of the acetyl group from histones by histone deacetylases (HDACs) is associated with gene silencing. Trichostatin A (TSA) was isolated from the metabolites of strains of *Streptomyces hygroscopicus* [10]. It is active against trichophytons and some fungi, and selectively inhibits enzymes of the class I and II HDAC families. TSA serves as an inhibitor of the eukaryotic cell cycle and inducer of morphological reversion in transformed cells. Aberrations in the transcription of key genes regulating vital cellular functions, such as cell-cycle regulation, proliferation, and apoptosis, are linked to the altered expression of HDACs; therefore, HDACs represent promising therapeutic targets for treating cancer. In contrast to conventional chemotherapy drugs, HDAC inhibitors provide high selectivity; several such inhibitors have been approved for the treatment of cutaneous T-cell lymphoma [11,12]. 

HDAC inhibitors appear suited to the treatment of bladder UC. In 76% of bladder tumors, inactivating mutations have been identified in at least one chromatin regulatory gene [13]. Additionally, an increase in the expression of HDACs was observed in high-grade UC [14]. Some HDAC inhibitors, such as TSA and belinostat, have demonstrated antitumor effects on UC cell lines through induced apoptosis and cell-cycle blockades [15,16,17,18]. Li et al. reported the synergistic effects of the HDAC inhibitor, AR-42, and cisplatin against bladder cancer [19]. In 2011, Yoon et al. reported the synergistic effects of TSA and resensitization to cisplatin treatment in human UC cells [20]. In a similar study, Yeh et al. demonstrated the resensitization of gemcitabine-resistant UC cells through TSA treatment [21]. 

The extracellular signal-regulated kinase (ERK) signaling pathway is a common downstream pathway of several growth factor receptor tyrosine kinases and is involved in the regulation of cell growth proliferation, survival, and apoptosis [22]. The rapidly accelerated fibrosarcoma (RAF) kinase/mitogen-activated protein kinase (MEK)/ERK pathway can govern drug resistance, apoptosis, and sensitivity to chemotherapy and targeted therapy and was reported to be a therapeutic target for cancer treatment [23]. RAF kinases are a family of serine threonine kinases that phosphorylate and activate MEK1/2, which then phosphorylates and activates ERK1/2. When activated, ERK1/2 phosphorylates various downstream substrates involved in multiple cellular responses—including cytoskeletal changes and gene transcription—in tumorigenesis [24]. In 2018, our team demonstrated a synergistic cytotoxicity induced through the RAF/MEK/ERK pathway in combination with TSA and three first-line chemotherapy drugs—cisplatin, gemcitabine, and doxorubicin—in UC cells based on clinical evidence [25]. The aforementioned studies suggest that HDAC inhibitors improve the therapeutic efficacy of chemotherapeutic drugs by overcoming resistance. Nonetheless, the mechanisms underlying the augmented cytotoxicity and resensitization of UC to drug treatment have yet to be elucidated.

In this in vitro and in vivo study, we sought to determine whether the ability of the antifungal antibiotic TSA to inhibit class I and class II HDAC enzyme families could enhance the efficacy of paclitaxel for treating human bladder UC. We also sought to identify the mechanism underlying the synergistic effects of paclitaxel chemotherapy when administered alongside TSA.

## 2. Results

### 2.1. TSA Enhances the Cytotoxicity of Paclitaxel and Reduces Viability in Human UC Cells

The effects of TSA alone and in combination with paclitaxel on the viability of UC cells were first assessed through MTT assay. As shown in Figure 1A, exposure to TSA or paclitaxel alone for 24 or 48 h reduced cell viability in a dose-dependent manner (0–1 μM) in BFTC-905 and BFTC-909 cells. Next, we evaluated the cytotoxic effects on UC cells following exposure to TSA (0.5 μM) combined with paclitaxel at various concentrations for 24 or 48 h. As shown in Figure 1B, TSA enhanced the cytotoxic effects of the chemotherapy drug, paclitaxel, toward BFTC-905 and BFTC-909 cells.

### 2.2. TSA Potentiates the Apoptotic Effect of Paclitaxel on UC Cells

The apoptotic effects on BFTC-905 and BFTC-909 cells of TSA alone and in combination with paclitaxel were evaluated through flow cytometry with annexin V and 7-AAD staining. After 24 h of exposure, TSA (0.5 µM) alone induced apoptosis in BFTC-905 and BFTC-909 cells. As shown in Figure 2A,B, TSA significantly potentiated the apoptotic effects of paclitaxel on UC cells. Additionally, Western blotting revealed that TSA combined with the chemotherapeutic agent increased the levels of cleaved caspase-3 and cleaved PARP compared with the chemotherapy drug alone (Figure 2C).

### 2.3. Paclitaxel in Combination with TSA Synergistically Inhibits Viability in Human UC Cells

We evaluated the synergistic effects of TSA in combination with paclitaxel at a concentration ratio of 10:1; the dose–effect plot and index–effect plot are illustrated in Figure 3. Experimental data related to the CI are listed in Table 1. The aforementioned combination consistently exhibited synergistic effects (CI < 1) in terms of reducing cell viability in BFTC-905 and BFTC-909 cells.

### 2.4. TSA Suppression of ERK Pathway Activation Following Paclitaxel Treatment in Human UC Cells

TSA and paclitaxel are two agents known to increase the levels of acetylated tubulin. We first examined whether the treatment of advanced UC cells with TSA and paclitaxel would simultaneously lead to a synergistic increase in acetylated tubulin and a consequent synergistic antitumor effect. As shown in Figure 4, cotreatment does not further enhance the levels of acetylated tubulin in advanced UC cells compared with TSA alone; this suggests that the increase of acetylated tubulin per se is not the underlying mechanism of the observed synergism. The ERK pathway—the downstream signaling pathway of growth factor receptor tyrosine kinases—is involved in the regulation of cell growth proliferation, apoptosis, and survival. The ERK pathway was demonstrated to govern drug resistance, apoptosis, and sensitivity to chemotherapy and targeted therapy; however, the role of the ERK pathway in cases of TSA combined with chemotherapy drugs has yet to be elucidated. Therefore, we examined the expression of downstream phospho-ERK1/2 following treatment with TSA alone, paclitaxel alone, and their combination. When administered alone, paclitaxel activated the ERK pathway. Treatment with 0.5 μM TSA in combination with paclitaxel suppressed the induction of phosopho-ERK1/2 by the chemotherapy drug (Figure 4).

### 2.5. Paclitaxel-Induced Antitumor Effects Were Enhanced by TSA in a Xenograft Mouse Model

A xenograft mouse model was used for in vivo analysis of the antitumor effects of chemotherapy, TSA alone, and a combination of both. The analysis involved injecting BFTC-905 and BFTC-909 cells mixed with Matrigel subcutaneously into the flanks of homozygous nude mice. As described in the Materials and Methods section, the mice were divided into four groups (*n* = 5 per group) and subsequently treated with paclitaxel, TSA, or their combination intraperitoneally; mice treated with normal saline were used as the control group. As shown in Figure 5, the combination of chemotherapy and TSA exerted considerably stronger antitumor effects on the BFTC-905 and BFTC-909 xenografts than did the chemotherapy drug or TSA alone. These results confirm our in vitro findings that the antitumor effects of TSA on UC can be improved through administration in combination with chemotherapy drugs.

## 3. Discussion

Cisplatin-based systemic chemotherapy remains the first-line treatment for patients with metastatic UC [26]. However, progression persists after cisplatin-based chemotherapy in 30–50% of advanced UC cases. Second-line chemotherapy drugs for advanced and metastatic UC have been developed, including taxanes and vinflunine [27,28,29]. Regarding the treatment of advanced UC, combination chemotherapy regimens have proven more effective than singular agents [30]. The most extensively studied second-line combination regimen is paclitaxel plus gemcitabine, which achieved an overall response rate of 30–70% [31]. However, many patients experience disease progression even after four to six cycles of second-line treatments.

To overcome the chemotherapy resistance of advanced or metastatic UC, the use of TSA as a combination agent to influence gene expression and enhance the antitumor effect of gemcitabine on advanced UC was explored [21]. Gemcitabine exhibits synergistic antitumor effects when combined with TSA to treat advanced UC. Paclitaxel is a toxin that stabilizes microtubules throughout cell cycles, thereby killing actively dividing cancer cells. Therefore, paclitaxel has been widely used in oncology, including for UC treatment [27,28]. In this study, we examined whether TSA—a potent inhibitor of HDAC6 as well as other HDACs—synergistically potentiates the antitumor effect of paclitaxel on advanced UC. Advanced UC undergoes more prominent apoptosis when treated simultaneously with paclitaxel and TSA than with either drug alone. In line with the in vitro culture results, the in vivo xenograft model clearly supported the hypothesis that TSA works synergistically with paclitaxel. Although the cooperative antitumor effect between TSA and paclitaxel was observed in papillary serous endometrial cancer cells [32], whether the synergistic effect holds for other cancer types, especially those that exhibit drug resistance, is unclear. Our study results indicate that the synergistic antitumor effect is well preserved in high-grade UC cells despite their insensitivity to other chemotherapeutic agents.

Because acetylated tubulin increased after treatment with TSA or paclitaxel, synergistic enrichment of acetylated tubulin is an attractive explanation for the observed synergistic killing of advanced UC through the combination of TSA and paclitaxel. Although synergistic enrichment of acetylated tubulin after treatment with TSA and paclitaxel has been noted in endometrial cancer cells, it has not been observed in advanced UC cells. The mechanisms underlying the synergistic antitumor effect of TSA and paclitaxel appear pleiotropic; patients with different cancer types may benefit from the combination of TSA and paclitaxel for different reasons. TSA is such a strong HDAC inhibitor that it also leads to histone hyperacetylation, and thus markedly reshapes the cancer cell transcriptome. TSA may resensitize chemoresistant cancer cells through microtubule-independent pathways. The Raf/MEK/ERK signaling pathway is involved in the regulation of cell growth proliferation, survival, and apoptosis [22]. This pathway is often up-regulated in cancer cells and serves as a promising anti-cancer target. In this study, we discovered that TSA effectively remodeled the aberrant ERK pathway found in advanced UC cells. After treatment with TSA and paclitaxel, the advanced UC cells did not exhibit cytoprotective activation of the ERK in the same manner as they had after treatment with paclitaxel alone. Consequently, advanced UC cells become more susceptible to apoptosis in vitro and tumor regression in vivo. Notably, TSA-mediated suppression of the ERK pathway also operated synergistically when combined with another cytotoxic agent, namely gemcitabine, in another chemoresistant UC model. This suggests that TSA is a general chemosensitizer that cooperates effectively with multiple chemotherapeutic agents. TSA-mediated suppression of the ERK pathway is particularly appealing because this pathway serves as a signaling hub that receives growth input from several tyrosine kinase receptors that are frequently aberrantly activated or overexpressed in human cancers [33]. For example, mutations in the RAS genes that activate the RAF/MEK/ERK pathway account for the greatest number of common mutations found in bladder cancer. Up to 13% of all UC harbors mutations of *HRAS*, *KRAS*, or *NRAS* [34]. As a proof-of-principle experiment, we demonstrated that TSA—an experimental HDAC inhibitor—is promising for the treatment of advanced UC. This finding is ready to be applied in clinical studies on humans. To date, several natural and synthetic HDAC inhibitors have been developed and are under investigation for clinical use. For example, the Food and Drug Administration has approved vorinostat and romidepsin with indications for refractory cutaneous T-cell lymphoma [35,36]. Some of these clinically approved HDAC inhibitors likely share antitumor properties with TSA.

Our study had some limitations. First, only one bladder UC cell line and one UTUC cell line were used to represent UC tumor behavior. Additionally, the xenograft mouse model may not have fully represented the cancer microenvironment. Second, we did not evaluate for any alterations of epigenetic modulation after TSA administration. Third, the cascade of the ERK pathway involves regulatory proteins, which give rise to multiple facets of tumor chemoresistance processes.

In conclusion, TSA synergistically enhances the cytotoxic effects of paclitaxel. One prominent mechanism is TSA-mediated suppression of the ERK pathway; this allows TSA to potentially cooperate with various chemotherapy drugs. The findings of this study forecast new avenues to circumvent the drug resistance observed in patients with advanced UC.

## 4. Material and Methods

### 4.1. Cell Culture

We used two human UC cell lines in this study. BFTC-905—a human urinary bladder UC cell line—was derived from a 51-year-old Taiwanese woman with a grade III papillary UC of the urinary bladder in 1990. BFTC-909 was derived from the sarcomatoid component of grade III UC of the renal pelvis exhibiting a fibroblastic growth pattern [37]. Both cell lines were purchased from the Food Industry Research and Development Institute, Hsinchu, Taiwan. The cell lines were cultured in Dulbecco’s modified Eagle’s medium supplemented with 10% (for BFTC-909) or 15% (for BFTC-905) fetal bovine serum, 100 U/mL penicillin, and 100 µg/mL streptomycin (Invitrogen, Carlsbad, CA, USA).

### 4.2. Reagents and Chemicals

TSA solution and a TSA compound were obtained from Sigma-Aldrich (St. Louis, MO, USA). The chemotherapeutic agent—paclitaxel—used in this study was from taken from clinical medication (Phyxol, Sinphar, Taiwan). All other chemicals were obtained from Sigma-Aldrich or Merck Millipore (Billerica, MA, USA).

### 4.3. Antibodies

For Western blotting, antibodies against caspase-8, cleaved caspase-3, cleaved poly (adenosine diphosphate–ribose) polymerase (PARP), phospho-ERK1/2, and acetyl-alpha-tubulin were purchased from Cell Signaling Technologies (Danvers, MA, USA). Antibodies against ERK1/2 and alpha-tubulin were purchased from Santa Cruz Biotechnology (Santa Cruz, CA, USA), and glyceraldehyde 3-phosphate dehydrogenase (GAPDH) antibodies were purchased from Genetex (Irvine, CA, USA).

### 4.4. Cell Viability Assay

This study used 3-(4,5-dimethylthiazol-2-yl)-2,5-diphenyltetrazolium (MTT; Sigma-Aldrich) to determine cell viability. Cells suspended in culture medium were seeded in 96-well microplates (2000–4500 cells/well) and incubated at 37 °C for 24 or 48 h before drug treatment. After exposure to drugs, the cells were incubated with 0.5 mg/mL MTT in complete medium at 37 °C for 4 h. Thereafter, dimethyl sulfoxide (DMSO) was applied to dissolve the reduced MTT crystals, and a plate reader (Thermo Scientific Multiskan GO, Thermo Scientific Pierce, Rockford, IL, USA) was used to measure the absorbance by the solvents at 540 nm.

### 4.5. Combination Index

The combined effects of the chemotherapeutic agents and TSA were determined using CalcuSyn software (version 1.1.1, Biosoft, Cambridge, UK). The combined effect at combination ratios of 10:1 (TSA to paclitaxel) was evaluated through median-effect and combination index (CI) analysis as described in previous studies [38,39]. The combined dose effects are described as the median, dose, and CI effects in Figure 3. CI values less than, equal to, and greater than 1 were defined as synergistic, additive, and antagonistic effects, respectively [38].

### 4.6. Western Blot Analysis

To determine protein expression, BFTC-905 and BFTC-909 cells were lysed with cell lysis buffer (Cell Signaling Technologies) on ice after being washed with cold phosphate-buffered saline (PBS). After centrifugation of cell lysates at 18,000× *g* for 10 min at 4 °C, the supernatants were collected, and bicinchoninic acid protein assay (Thermo Scientific Pierce) was used to detect the total protein concentration. Equal amounts of proteins, which were mixed with TOOLS sample loading buffer (Biotools, Taipei, Taiwan), from each cell line were subjected to sodium dodecyl sulfate-polyacrylamide gel electrophoresis and then transferred onto polyvinylidene fluoride membranes (Merck Millipore). After being blocked with 5% bovine serum albumin (BSA) in PBS, the membranes were incubated with various primary antibodies in PBS at 4 °C overnight. After two washes with Tris-buffered saline containing 0.05% Tween 20 (TBST), the membranes were incubated with horseradish peroxidase-conjugated secondary antibodies (Genetex) at recommended dilution ratios in PBS at room temperature for 2 h. The antibody-marked membranes were again washed twice with TBST and visualized using enhanced chemiluminescence substrates (Merck Millipore) in an ImageQuant LAS 4000 (GE Healthcare) system. Experiments were performed in triplicate.

### 4.7. Apoptosis Assay

The apoptosis assay was performed according to the manufacturer’s protocol using Muse Annexin V and a Dead Cell Kit (Merck Millipore). Cell lines were preincubated with relative drugs or in controlled conditions for 24 h and then trypsinized for centrifugation. Cells were centrifuged at 100× *g* and washed once with PBS. After being washed, the cells were resuspended in 2–20% BSA solution and subsequently incubated with an equal volume of Muse Annexin V and Dead Cell Reagent for 20 min at room temperature to stain the apoptotic markers, annexin V and 7-AAD, that had translocated to the outer membrane. The stained apoptotic cells were then examined and quantified through flow cytometry using a Muse Cell Analyzer (Merck Millipore).

### 4.8. In Vivo Xenograft

All animal care and experimental procedures were conducted in accordance with the protocols approved by the Institutional Animal Care and Use Committee of the College of Public Health and College of Medicine at National Taiwan University. Studies involving animals followed the ARRIVE guidelines for reporting experiments concerning animals. In the present study, 40 animals were used. BFTC-905 or BFTC-909 cells (5 × 10^5^ to 1 × 10^6^ cells) were suspended in 200 μL of serum-free medium and mixed with an equivalent volume of Matrigel (BD Biosciences). Eight-week-old nude mice (obtained from the Taiwan National Laboratory Animal Center, Taipei, Taiwan) were injected subcutaneously with the mixtures into the dorsal flanks. After the tumors had grown to 100–150 mm^3^, the mice were treated with paclitaxel alone, paclitaxel combined with TSA, or TSA alone; treatment was performed in parallel with control group mice (*n* = 5 for each group). Paclitaxel (2 mg/kg, three times weekly) or TSA (1 mg/kg, three times weekly) in normal saline was intraperitoneally injected into the chemotherapy- and TSA-treated mice, respectively, for 4 weeks. Meanwhile, the combination group received the same doses of the drug and TSA, which were administered at the same frequency over the same duration. Mice receiving a mixture of DMSO and normal saline were designated as the untreated control group.

The tumor sizes were measured by calipers every 4 days and calculated as volume = π/6[(*A* + *B*)/2]^3^; *A* is the longest diameter of the tumor and is perpendicular to the tumor diameter, *B*. After 4 weeks of treatment, the tumors were abscised and photographed before being frozen in liquid nitrogen and stored at −80 °C.

### 4.9. Statistical Analyses

Statistical analyses were performed using GraphPad Prism 7 software with all data presented as mean ± standard deviation (SD) or standard error of the mean (SEM) and analyzed using one-way ANOVA; *p* < 0.05 was considered statistically significant.

## Figures and Tables

**Figure 1 ijms-20-01162-f001:**
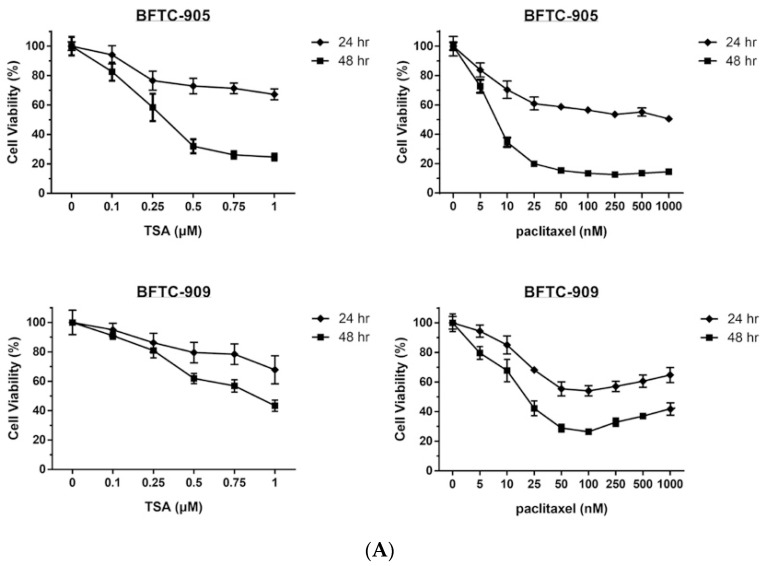
MTT cell viability assay of BFTC-905 and BFTC-909 with trichostatin A (TSA), paclitaxel, or combination treatment. Cells were treated with the relevant drug for 24 h, and cell viability was analyzed at OD540 with untreated cells defined as 100% (*n* = 6). Cell viability is presented as mean ± SD. (**A**) The 50% inhibitory concentration (IC_50_) of TSA treatment at 24 and 48 h was >1µM and =27 nM, respectively, for BFTC-905 and >1µM and =88 nM, respectively, for BFTC-909. By contrast, the IC_50_ of paclitaxel treatment were approximately 1 µM and 7.5 nM, respectively, for BFTC-905 and approximately 0.05 µM and 1.7 nM, respectively, for BFTC-909 at 24 and 48 h. (**B**) Cotreatment with paclitaxel and TSA resulted in a significant difference in cell viability compared with paclitaxel alone (one-way ANOVA). * *p* < 0.05.

**Figure 2 ijms-20-01162-f002:**
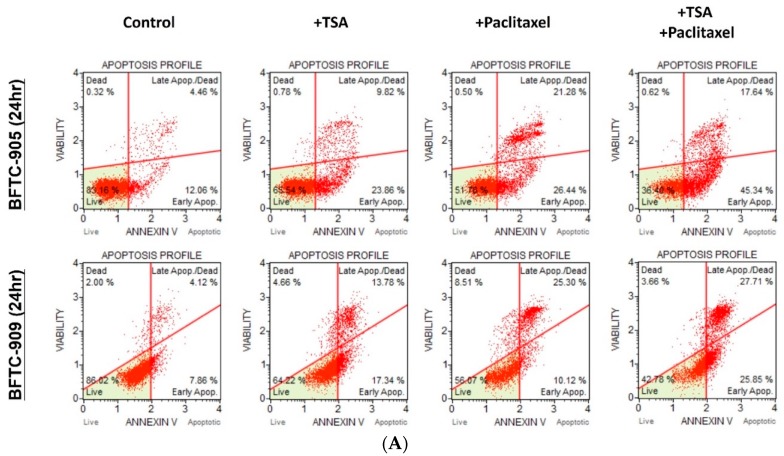
Flow cytometry analysis and apoptosis signaling transduction with TSA, paclitaxel treatment. Cells were stained with annexin V and 7-AAD for the analysis of apoptosis through flow cytometry (*n* = 3), and apoptosis pathway analysis was conducted through Western blotting. Total apoptosis values are presented as mean ± SD. (**A**,**B**) Cotreatment of TSA and paclitaxel induced significant apoptosis in both UC cell lines compared with either treatment alone (one-way ANOVA). (**C**) Paclitaxel treatment plus TSA reduced pro-caspase-8 and enhanced both cleaved-form molecules in BFTC-905 and BFTC-909. * *p* < 0.05.

**Figure 3 ijms-20-01162-f003:**
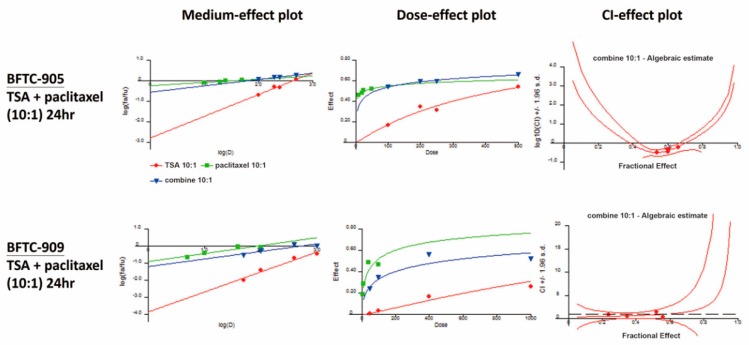
Combination indexes (CIs). Synergistic effects were confirmed through cell viability assay (*n* = 4), and the results were calculated in CalcuSyn. TSA and paclitaxel were at a 10:1 concentration ratio. CIs are presented in the CI–effect plot (as log_10_CI ± SD). CI values <1, =1, and >1 were defined as synergistic, additive, and antagonistic effects, respectively.

**Figure 4 ijms-20-01162-f004:**
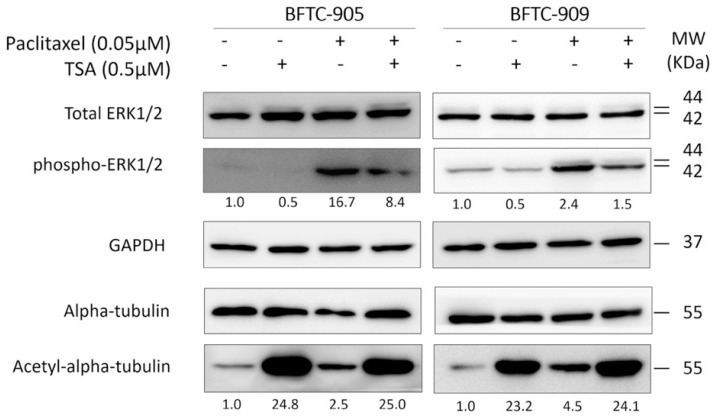
Western blotting for the ERK pathway and acetylation. Cells were treated with 0.5 μM TSA or 0.05 μM paclitaxel for 24 h, and cell lysates were collected for further Western blotting. An increase of phosphor-ERK1/2 suggested a potential drug-resistant phenomenon in the paclitaxel-only group; however, phosphor-ERK1/2 declined in the TSA alone and combined groups, suggesting a potential antiresistance-like effect. To confirm acetylation under the TSA treatment, acetyl-alpha-tubulin served as an acetylation marker. In both the TSA and combination groups, acetyl-alpha-tubulin increased robustly, indicating the effect of the histone deacetylation inhibitor.

**Figure 5 ijms-20-01162-f005:**
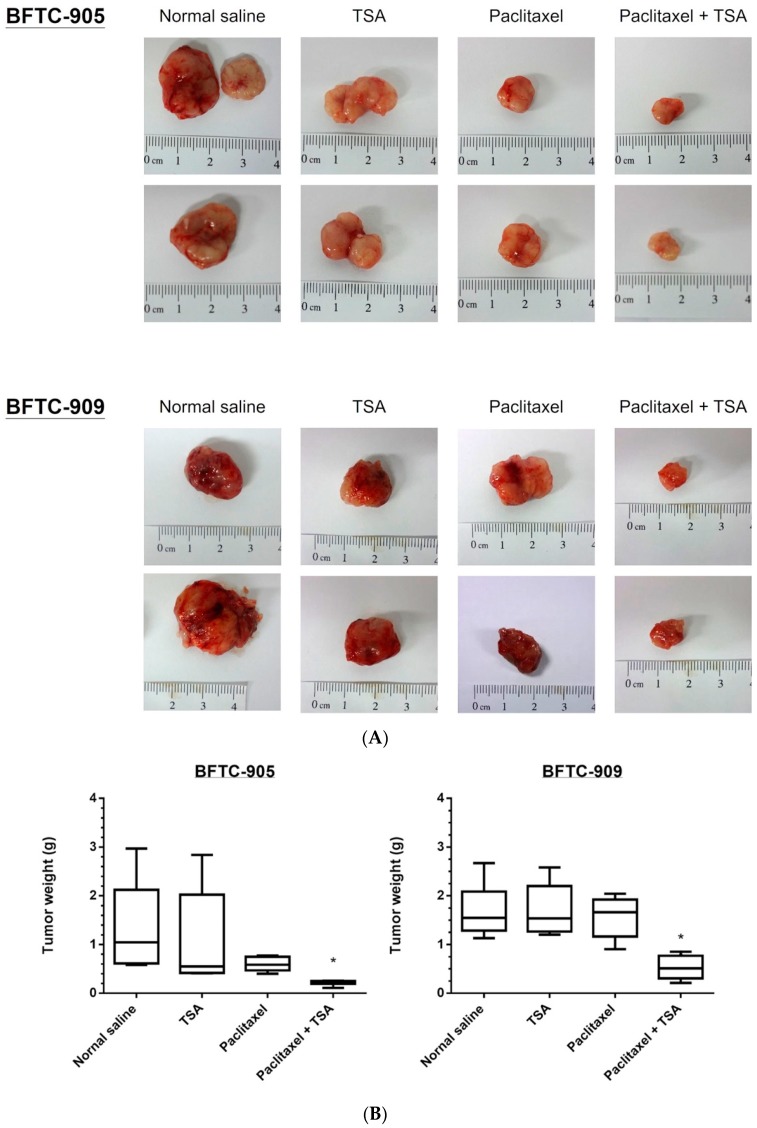
BFTC-905 and BFTC-909 xenograft in nude mice under TSA and paclitaxel cotreatment revealed a significant inhibition of tumor volume growth. The lateral sides of the nude mice 6–8 weeks old were xenografted with BFTC-905 or BFTC-909 cells (*n* = 5 mice per group). Drug treatments were administered three times each week for a month, starting 2 weeks after xenograft (1 mg TSA/kg, 2 mg paclitaxel/kg, intraperitoneally). (**A**) Tumors after 4 weeks of treatment, (**B**) tumor weight progression over 4 weeks of treatment, and (**C**) tumor volume progression over 4 weeks of treatment (*n* = 6 tumors, data are presented as mean ± SEM). Tumor volumes show significant differences between the control and combined groups at week 4 (one-way ANOVA). * *p* < 0.05.

**Table 1 ijms-20-01162-t001:** CI. Combination consistently exhibited synergistic effects (CI < 1) in terms of reducing viability in BFTC-905 and BFTC-909 cells.

**BFTC-905 (24 h)**	**Paclitaxel (nM)**	**Fraction Affected (Fa)**	**TSA:Paclitaxel = 10:1**
**TSA (nM)**			**Combination Index (CI)**
100	10	0.54	0.330
200	20	0.60	0.369
250	25	0.60	0.461
500	50	0.66	0.640
**BFTC-909 (24 h)**	**Paclitaxel (nM)**	**Fraction Affected (Fa)**	**TSA:Paclitaxel = 10:1**
**TSA (nM)**			**Combination Index (CI)**
50	5	0.24	0.798
100	10	0.35	0.552
400	40	0.56	0.460
1000	100	0.52	1.516

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
