# Peer review of "Histone Deacetylase Inhibitor, Trichostatin A, Synergistically Enhances Paclitaxel-Induced Cytotoxicity in Urothelial Carcinoma Cells by Suppressing the ERK Pathway"

_ijms, 2019, doi:10.3390/ijms20051162_

Round 1
Reviewer 1 Report
The authors have succinctly expressed the limitations of this study. "First, only one bladder UC cell line and one UTUC cell line 335 were used as substitutes for the study of UC tumor behavior. In addition, the xenograft mouse model 336 may not have fully represented the cancer microenvironment. Second, we did not evaluate the 337 alterations of epigenetic modulation after TSA administration. Third, the cascade of the ERK pathway 338 involves regulatory proteins, which give rise to multiple facets of tumor chemoresistance processes." In addition, the mechanism inferred is very limited in this age of next gen sequencing. None of the WB show the size of the proteins.
Author Response
Response:
Thanks for your precious comments. We have added molecular weight of proteins in the western blot.
Reviewer 2 Report
This is a well written paper that presents an interesting finding and the relevant data are clearly presented. I have several comments, some more significant than others.
The title is a little misleading. "via the ERK pathway" should be replaced with "by suppressing chemotherapy induced inactivation of the ERK pathway" or simply "by suppressing the ERK pathway."
In the last panel of Figure 1B, the percent cell viability of BFTC-909 cells after 48 hr of exposure to the highest concentration of paclitaxel in combination with TSA is actually greater than the percent cell viability after 48 hr of exposure to the same concentration of paclitaxel alone. The authors should address this result, which is at odds with their hypothesis. Does this mean that the synergism disappears with time when BFTB-909 cells are used? What would happen after 72 hours?
A preferred formula for calculating tumor volume is π/6 x [(A + B)/2] 3 .
The key and axis titles in Figure 3 are too small to be legible.
In lines 131 and 148, rpm should be converted to g.
In line 142, the sentence that begins with "Experiments are did..." needs to be re-written.
In line 145 "determined" should be changed to "performed."
In line 205, "Figure 2B" should be "Figure 2C."
In line 315, did the authors really mean to say "screwed ERK pathway??"
Finally, while the authors present a clear description of HDAC inhibitor function in the introduction, the connection between inhibition of histone deacetylation and suppression of the ERK pathway is unclear. This paper would be greatly strengthened if an experiment were added to show how inhibition of histone deacetylation leads to suppression of the ERK pathway, which, in turn, leads to synergism when TSA is combined with paclitaxel. Or does ERK suppression have nothing to do with TSA's HDAC inhibitor function? Either way, especially since these authors have already shown that TSA has similar synergism with at least once other chemotherapeutic agent besides paclitaxel, this important mechanistic point should be addressed.
Author Response
Reviewer 2
This is a well written paper that presents an interesting finding and the relevant data are clearly presented. I have several comments, some more significant than others.
Response:
1. We have revised the title as “Histone deacetylase inhibitor trichostatin A synergistically enhances paclitaxel-induced cytotoxicity in urothelial carcinoma cells by suppressing the ERK pathway”
2. We wholly agree with your comments. In Figure 1B, it seemed that TSA did not enhance cytotoxicity of paclitaxel on BFTC-909 cells at 48-hr exposure. The underlying causes to explain the phenomenon could be complex and undetermined. The cell viability after 72-hour treatment with paclitaxel and TSA could provide some information. We apologized that we failed to performed the experiment within 10-day submission deadline. Nevertheless, synergism and enhancement are not totally same. Synergism is “mutual” whereas enhancement is “one-sided”. Synergism needs to be determined by calculating drug combination index (CI) as we showed in Figure 3, which actually indicated the synergism of TSA and paclitaxel on BFTC-905 and BFTC-909 cells.
3. We have re-calculated the tumor volume and re-generated Figure 5C using the formula, π/6 x [(A + B)/2] 3. We also revised the descriptions of tumor volume calculation in Material and Methods[B1] (Line 178-179).
4. We have amended Figure 3 using larger font size to make it legible.
5. (In lines 131 and 148, rpm should be converted to g.)We have amended rpm to g as you suggested. (line 141 and 157)
6. (In line 142, the sentence that begins with "Experiments are did..." needs to be re-written.)We have re-written the sentence as you suggested. (line 152)
7. (In line 145 "determined" should be changed to "performed.")We have corrected this in accordance with your comments. (line 154)
8. (In line 205, "Figure 2B" should be "Figure 2C.")We have corrected the error. (line 215)
9. (In line 315, did the authors really mean to say "screwed ERK pathway??")We replaced “screwed” as “aberrant” and revised the description as follows: “we discovered that TSA effectively remodeled the aberrant ERK pathway found in advanced UC cells.” (line 328-329)
10.(Finally, while the authors present a clear description of HDAC inhibitor function in the introduction, the connection between inhibition of histone deacetylation and suppression of the ERK pathway is unclear. This paper would be greatly strengthened if an experiment were added to show how inhibition of histone deacetylation leads to suppression of the ERK pathway, which, in turn, leads to synergism when TSA is combined with paclitaxel. Or does ERK suppression have nothing to do with TSA's HDAC inhibitor function? Either way, especially since these authors have already shown that TSA has similar synergism with at least once other chemotherapeutic agent besides paclitaxel, this important mechanistic point should be addressed.)
We wholly agree with your comments. To clarify the points you raised, we used PD98059, a specific MEK inhibitor, in combination with paclitaxel and examined the cell viability of UC cells. The figures are embedded as follows. We found that PD98059 decreased paclitaxel-induced phospho-ERK1/2 activation. Inhibition of ERK pathway by PD98059 enhanced pactlitaxel-induced cytotoxicity in UC cells.
[B1]Please note that this is listed after the Discussion section per IJMSguidelines.

Reviewer 3 Report
Hsu et al., present their findings on synergistic interactions between TSA and Paclitaxel to combat drug resistant urothelial carcinoma in their paper ‘Histone deacetylase inhibitor trichostatin A synergistically enhances paclitaxel-induced cytotoxicity in urothelial carcinoma cells via the ERK pathway’. They present their data highlighting how TSA in combination with paclitaxel reduces overall cancer cell viability and increases apoptosis, and substantiate their findings in vivo. Their proposed mechanism of action suggests that pERK upregulation upon paclitaxel exposure can be reduced/inhibited by TSA to achieve a more potent anti-tumor effect. Although the background and discussion are very well written and the findings are novel and robust, a few additional experiments will significantly strengthen the paper.
Major Concerns:
1. Given TSA is known to affect HDACs, the authors have not shown the effect of TSA on inhibiting/altering HDAC levels. This needs to be shown experimentally.
2. Furthermore, the connection between how a HDAC inhibitor could reduce pERK levels is unclear. Either further experiments or discussion should be incorporated to address this issue.
3. Thirdly, given the authors claim that TSA increases tumor inhibition by reducing pERK upon paclitaxel treatment, it would be prudent to inhibit pERK using a ERK or MEK inhibitor (in paclitaxel treated cells) to show that the TSA is indeed functioning through repression of ERK pathway.
Minor Concerns:
1. In Figure 1&2, the authors employ t-tests, and make grouped comparisions. The correct statistic test to use in this instance is an ANOVA.
2. in Figure 2 the Total caspase-3 levels, and PARP levels must be shown. Full length blots must be made available, and the molecular weight of proteins must be indicated in the WBs.
3. In Figure 5B, is there no Significance in the in vivo results even when comparing the TSA+paclitaxel group to control? (Anova must be used again). If not, can the authors comment on the discrepancy between their in vitro and in vivo observations?
4. Include more relevant literature citing role of ERK pathway in cancer
Author Response
Reviewer 3
Major Concerns:
1. Given TSA is known to affect HDACs, the authors have not shown the effect of TSA on inhibiting/altering HDAC levels. This needs to be shown experimentally.
Response:
We wholly agree with your comments. We fail to show the effect of TSA on altering HDAC level within 10-day submission deadline. Nevertheless, TSA has been a well-known pan-inhibitor of HDACs in previous studies. The cellular response secondary to HDAC inhibition is complex. We also revised the discussion on this issue as follows:”TSA is such a strong HDAC inhibitor that it also leads to histone hyperacetylation and thus markedly reshapes the cancer cell transcriptome. In this study, we discovered that TSA effectively remodeled the aberrant ERK pathwayfound in advanced UC cells. After treatment with TSA and paclitaxel, the advanced UC cells did not exhibit cytoprotective activation of ERK in the same manner as they had after treatment with paclitaxel alone. Consequently, advanced UC cells become more susceptible to apoptosis in vitro and tumor regression in vivo.”(Line 326-334)
2. Furthermore, the connection between how a HDAC inhibitor could reduce pERK levels is unclear. Either further experiments or discussion should be incorporated to address this issue.
Response:
As we shown in Figure 4, TSA (0.5uM) alone suppressed phospho-ERK1/2 without changing total ERK 1/2 in both UC cells. We have revised the discussion as follows:” TSA is such a strong HDAC inhibitor that it also leads to histone hyperacetylation and thus markedly reshapes the cancer cell transcriptome. In this study, we discovered that TSA effectively remodeled the aberrant ERK pathway found in advanced UC cells. After treatment with TSA and paclitaxel, the advanced UC cells did not exhibit cytoprotective activation of ERK in the same manner as they had after treatment with paclitaxel alone. Consequently, advanced UC cells become more susceptible to apoptosis in vitro and tumor regression in vivo.”(Line 326-334)
3. Thirdly, given the authors claim that TSA increases tumor inhibition by reducing pERK upon paclitaxel treatment, it would be prudent to inhibit pERK using a ERK or MEK inhibitor (in paclitaxel treated cells) to show that the TSA is indeed functioning through repression of ERK pathway.
Response:
We wholly agree with your comments. To clarify the points you raised, we used PD98059, a specific MEK inhibitor, in combination with paclitaxel and examined the cell viability of UC cells. The figure is embedded as follows. We found that PD98059 decreased paclitaxel-induced phospho-ERK1/2 activation. Inhibition of Raf/MEK/ERK pathway by PD98059 enhanced pactlitaxel-induced cytotoxicity in UC cells.
Minor Concerns:
1. In Figure 1&2, the authors employ t-tests, and make grouped comparisions. The correct statistic test to use in this instance is an ANOVA.
Response:
We have re-analyzed the data of Figure 1&2 using ANOVA and renew the figure.
2. in Figure 2 the Total caspase-3 levels, and PARP levels must be shown. Full length blots must be made available, and the molecular weight of proteins must be indicated in the WBs.
Response:
We have added molecular weight of proteins in the western blot.
3. In Figure 5B, is there no Significance in the in vivo results even when comparing the TSA+paclitaxel group to control? (Anova must be used again). If not, can the authors comment on the discrepancy between their in vitro and in vivo observations?
Response:
We have re-analyzed differences of tumor weight between four groups (control, TSA, pactlitaxel, paclitaxel and TSA) using ANOVA.
4. Include more relevant literature citing role of ERK pathway in cancer
Response:
We have added the following description in introduction (Line 82-90) and discussion:”Raf/MEK/ERK signaling pathway is involved in the regulation of cell growth proliferation, survival, and apoptosis. This pathway is often up-regulated in cancer cells and served as a promising anti-cancer target. “(Line 328-330)
Round 2
Reviewer 3 Report
Not all concerns have been answered, but manuscript has atleast been partially improved.